# CO$_2$ directly modulates connexin 26 by formation of carbamate bridges between subunits

Louise Meigh[1], Sophie A Greenhalgh[1], Thomas L Rodgers[2,3], Martin J Cann[3,4], David I Roper[1], Nicholas Dale[1]*

[1]School of Life Sciences, University of Warwick, Coventry, United Kingdom; [2]Biophysical Sciences Institute, University of Durham, Durham, United Kingdom; [3]Department of Chemistry, University of Durham, Durham, United Kingdom; [4]School of Biological and Biomedical Sciences, University of Durham, Durham, United Kingdom

**Abstract** Homeostatic regulation of the partial pressure of CO$_2$ (PCO$_2$) is vital for life. Sensing of pH has been proposed as a sufficient proxy for determination of PCO$_2$ and direct CO$_2$-sensing largely discounted. Here we show that connexin 26 (Cx26) hemichannels, causally linked to respiratory chemosensitivity, are directly modulated by CO$_2$. A 'carbamylation motif', present in CO$_2$-sensitive connexins (Cx26, Cx30, Cx32) but absent from a CO$_2$-insensitive connexin (Cx31), comprises Lys125 and four further amino acids that orient Lys125 towards Arg104 of the adjacent subunit of the connexin hexamer. Introducing the carbamylation motif into Cx31 created a mutant hemichannel (mCx31) that was opened by increases in PCO$_2$. Mutation of the carbamylation motif in Cx26 and mCx31 destroyed CO$_2$ sensitivity. Course-grained computational modelling of Cx26 demonstrated that the proposed carbamate bridge between Lys125 and Arg104 biases the hemichannel to the open state. Carbamylation of Cx26 introduces a new transduction principle for physiological sensing of CO$_2$.

*For correspondence: n.e.dale@ warwick.ac.uk

Competing interests: The authors declare that no competing interests exist.

## Introduction

CO$_2$ is the unavoidable by-product of cellular metabolism. Humans produce approximately 20 moles of CO$_2$ per day (*Marshall and Bangert, 2008*). The dissolved CO$_2$ can readily combine with water, aided by carbonic anhydrase, to form H$_2$CO$_3$, which dissociates rapidly to H$^+$ and HCO$_3^-$. In any physiological solution therefore, the partial pressure of CO$_2$ (PCO$_2$) will be in equilibrium with, and inescapably related to, the pH and the concentration of HCO$_3^-$ of that solution. Regulation of PCO$_2$ is thus a vital homeostatic function that is linked to acid-base balance.

As might be expected, chemosensory reflexes regulate the frequency and depth of breathing to ensure homeostatic control of blood gases. The field of respiratory chemosensitivity has been dominated by 'reaction theory' which posits that pH is a sufficient signal for detection of changes in PCO$_2$ (*Loeschcke, 1982*). Many investigators therefore equate pH-sensing with CO$_2$-sensing. There are several areas of the medulla oblongata which contain neurons that respond to changes in pH/CO$_2$, especially near the highly vascularised ventral surface. For example a population of neurons highly sensitive to pH/CO$_2$ have been described in the retrotrapezoid nucleus (RTN) (*Mulkey et al., 2004, 2006*; *Guyenet et al., 2008*) and the medullary raphé nucleus (*Richerson, 2004*; *Ray et al., 2011*). Despite the acceptance of pH-sensing as the predominant mechanism by which PCO$_2$ is measured, there is substantial evidence for an additional and independent effect of molecular CO$_2$ (*Eldridge et al., 1985*; *Shams, 1985*; *Huckstepp and Dale, 2011*). For example, if pH is carefully controlled at the medullary surface, an increase in PCO$_2$ at constant pH will still enhance breathing by as much as a pH change at constant PCO$_2$ (*Shams, 1985*). We have recently shown that connexin 26 (Cx26)

**eLife digest** A number of gaseous molecules, including nitric oxide and carbon monoxide, play important roles in many cellular processes by acting as signalling molecules. Surprisingly, however, it has long been assumed that carbon dioxide – a gaseous molecule that is produced during cellular metabolism – is not a signalling molecule.

Controlling the concentration of carbon dioxide ($CO_2$) in a biological system is essential to sustain life, and it was thought that the body used pH – which is the concentration of hydrogen ions – as a proxy for the level of $CO_2$. The concentration of $CO_2$ is related to pH because $CO_2$ reacts with water to form carbonic acid, which quickly breaks down to form hydrogen ions and bicarbonate ions. This close relationship has led many researchers to equate pH-sensing with $CO_2$-sensing, and to suggest that a physiological receptor for $CO_2$ does not exist.

Recent research into structures called connexin hemichannels has challenged this view. Researchers found that when pH levels were held constant, increasing the level of $CO_2$ caused the structures to open up, suggesting that $CO_2$ could be directly detected by the hemichannels. Each hemichannel contains six connexin subunits, but the details of how the $CO_2$ molecules interact with the individual connexin subunits to open up the hemichannels remained mysterious.

Now Meigh et al. show that $CO_2$ molecules bind to a specific amino acid (lysine) at a particular place (residue 125) in one of the connexin subunits to form a carbamate group. This group then interacts with the amino acid (arginine) at residue 104 in a neighbouring connexin subunit to form a carbamate bridge between the two subunits. This leads to structural changes that cause the gap junction hemichannels to open and release signals that can activate other cells. Since connexin hemichannels are found throughout the human body, these results suggest that $CO_2$ might act as a signalling molecule in processes as diverse as the control of blood flow, breathing, hearing and reproduction.

hemichannels, open in response to increases in $PCO_2$ at constant extracellular pH and are an important conduit for the $CO_2$-dependent, as opposed to pH-dependent, release of ATP (*Huckstepp et al., 2010a*). Cx26 hemichannels contribute to the chemosensory control of breathing (*Huckstepp et al., 2010b*; *Wenker et al., 2012*). Hemichannels of two closely related connexins, Cx30 and Cx32, also exhibited $CO_2$-sensitive opening (*Huckstepp et al., 2010a*). Despite this evidence, widespread acceptance of direct sensing of $CO_2$ requires a detailed molecular explanation of any putative transduction system.

One possible way that $CO_2$ can interact with proteins is via carbamylation—the formation of a covalent bond between the carbon of $CO_2$ and a primary amine group. For example, $CO_2$ forms carbamate bonds with haemoglobin (*Kilmartin and Rossi-Bernardi, 1971*) and the plant enzyme RuBisCo (*Lundqvist and Schneider, 1991*). Here we document an important new advance—the mechanism by which $CO_2$ binds directly to Cx26, most probably via carbamylation of a lysine residue, to cause hemichannel opening. Our work establishes a new field of direct $CO_2$ sensing that can be mediated by $CO_2$-dependent carbamylation of certain β connexins. As these are widely distributed in the brain and other tissues, it is likely that direct detection of $CO_2$ via this mechanism is important in many different physiological processes.

## Results

We have previously demonstrated that Cx26, and two related β connexins, Cx30 and Cx32, open when exposed to modest increases in $PCO_2$ at constant pH (*Huckstepp et al., 2010a*). This previous study demonstrated, in inside-out and outside-out excised membrane patches at a transmembrane potential of −40 mV, that Cx26 hemichannel gating respectively increased and decreased in response to increases and decreases of $PCO_2$. To reconfirm our previous findings that Cx26, and not some other hemichannel senses $CO_2$ (*Huckstepp et al., 2010a*), we demonstrated that the $CO_2$-dependent dye loading of HeLa cells expressing Cx26 was blocked by 100 µM carbenoxolone, but unaffected by 1 mM probenecid, a blocker of pannexin-1, (*Silverman et al., 2008*), and 20 µM ruthenium red, a blocker of calhm1 (*Taruno et al., 2013*), (*Figure 1—figure supplement 1*). Parental HeLa cells do not exhibit $CO_2$-dependent dye loading demonstrating that the heterologous expression of Cx26 is essential for this function (*Huckstepp et al., 2010a*) (*Figure 3—figure supplement 1*).

## The extent of $CO_2$ chemosensitivity within the β connexins

To document the extent to which this sensitivity to $CO_2$ is limited within the β connexin family (*Figure 1F*), and to form the basis of a bioinformatic comparison to identify possible $CO_2$ binding motifs, we investigated whether another β connexin, Cx31, might also be sensitive to $CO_2$. We expressed, in HeLa cells, either rat Cx31 or rat Cx26 tagged at the C-terminal with mCherry and used a previously described dye loading assay (*Huckstepp et al., 2010a*) to test whether the cells could load with carboxyfluorescein (CBF) in a $CO_2$-dependent manner. As expected from our previous work, HeLa cells expressing the Cx26 readily loaded with CBF when exposed to this dye in the presence of elevated $PCO_2$ (55 mmHg, at pH 7.5, *Figure 1A,B*). However, HeLa cells expressing Cx31 failed to dye load in a $CO_2$-dependent manner (*Figure 1A,B*). As the connexins were tagged with mCherry, we could verify the presence of fluorescent puncta in both the Cx26 and Cx31 expressing HeLa cells (*Figure 1—figure supplement 2*). To check for the existence of functional hemichannels in the Cx31-expressing HeLa cells, we removed extracellular $Ca^{2+}$ as a positive control. This manipulation will open all types of connexin hemichannel. Parental HeLa cells do not load with dye when $Ca^{2+}$ is removed from the medium (*Figure 3—figure supplement 1*); they therefore possess virtually no endogenous hemichannels. The removal of extracellular $Ca^{2+}$ readily caused loading of CBF into the Cx31-expressing HeLa cells (*Figure 1A*, inset), demonstrating the presence of functional Cx31 hemichannels.

## Identification of a carbamylation motif in $CO_2$ sensitive β connexins

The $CO_2$-sensitivity in the β connexins therefore appears to be limited to the three closely related connexins, Cx26, Cx30 and Cx32, and Cx31 has no sensitivity to increases in $PCO_2$ (*Figure 1F*). We hypothesized that $CO_2$ carbamylated a lysine residue in Cx26 to induce conformational change and hence opening of the hemichannel. We therefore compared the sequences of the three $CO_2$-sensitive connexins to Cx31 to identify a lysine present in all three $CO_2$ sensitive connexins but absent from Cx31 (*Figure 1C*). This analysis revealed K125 and four further amino acids as forming a motif that was absent from Cx31. The existing crystal structure for Cx26 (*Maeda et al., 2009*), shows that K125 is at the end of an alpha helix and that the sequence KVREI ('carbamylation motif') orients K125 towards R104 on the neighbouring subunit (*Figure 1D*). The distance from K125 to R104 is only 6.5 Å (*Maeda et al., 2009*), strongly suggesting that if K125 were to be carbamylated it could form a salt bridge between these two residues in adjacent subunits (*Figure 1E*). Interestingly, R104 is present in Cx30, but conservatively substituted by a lysine residue in Cx32 (*Figure 1C*), which has a lower sensitivity to $CO_2$ than Cx26 (*Huckstepp et al., 2010a*).

## Insertion of the carbamylation motif into Cx31 creates a $CO_2$-sensitive mutant hemichannel

Our analysis predicts that if we were to introduce the putative carbamylation-motif of Cx26 into Cx31, the resulting mutant Cx31 (mCx31) should be sensitive to $CO_2$ as the lysine introduced into the sequence should be able to form a salt bridge with the native residue K104 in mCx31 once carbamylated (*Figure 1C–E and 2A*). We therefore made mCx31 in which the motif TQKVREI was introduced in place of K123H124 of the native connexin (*Figure 2A*). This insertion/substitution maintained the correct orientation of the K125 with respect to K104 of Cx31. HeLa cells expressing mCx31 displayed clear $CO_2$-dependent dye loading (*Figure 2B,C*). We confirmed the $CO_2$ sensitivity of mCx31 expressing HeLa cells by performing whole cell patch camp recordings. mCx31-expressing cells exhibited a conductance change of 3.3 ± 0.84 nS (mean ± SEM, n = 8, *Supplementary file 1*) when exposed to elevated $PCO_2$ (*Figure 2D*). Cells, expressing wild type Cx31 showed no $CO_2$-dependent changes in their whole cell conductance (mean conductance change −0.002 ± 0.023 nS, n = 6, *Supplementary file 1*, *Figure 2D*).

## K125 and R104 are essential for $CO_2$ sensitivity

To demonstrate that K125 is the key residue for interaction with $CO_2$, we made mCx31$^{K125R}$, by inserting TQRVREI into Cx31 in place of K123H124. Unlike lysine, the arginine side chain cannot be carbamylated by $CO_2$ as its pKa (12.5) is much higher than that of lysine (10.5), therefore this variant should have no sensitivity to $CO_2$. mCx31$^{K125R}$ did indeed lack sensitivity to $CO_2$ (*Figure 3A–C*, *Figure 3—source data 1*). This was not because the mutant channel failed to assemble correctly, as the positive control of zero $Ca^{2+}$ dye loading demonstrated the presence of functional hemichannels (*Figure 3A*, *Figure 3—figure supplement 1*). Next we investigated the relevant residues in Cx26 itself. The carbamate bridge that we propose must involve K125 (being carbamylated) and R104 (forming the salt bridge with the carbamylated

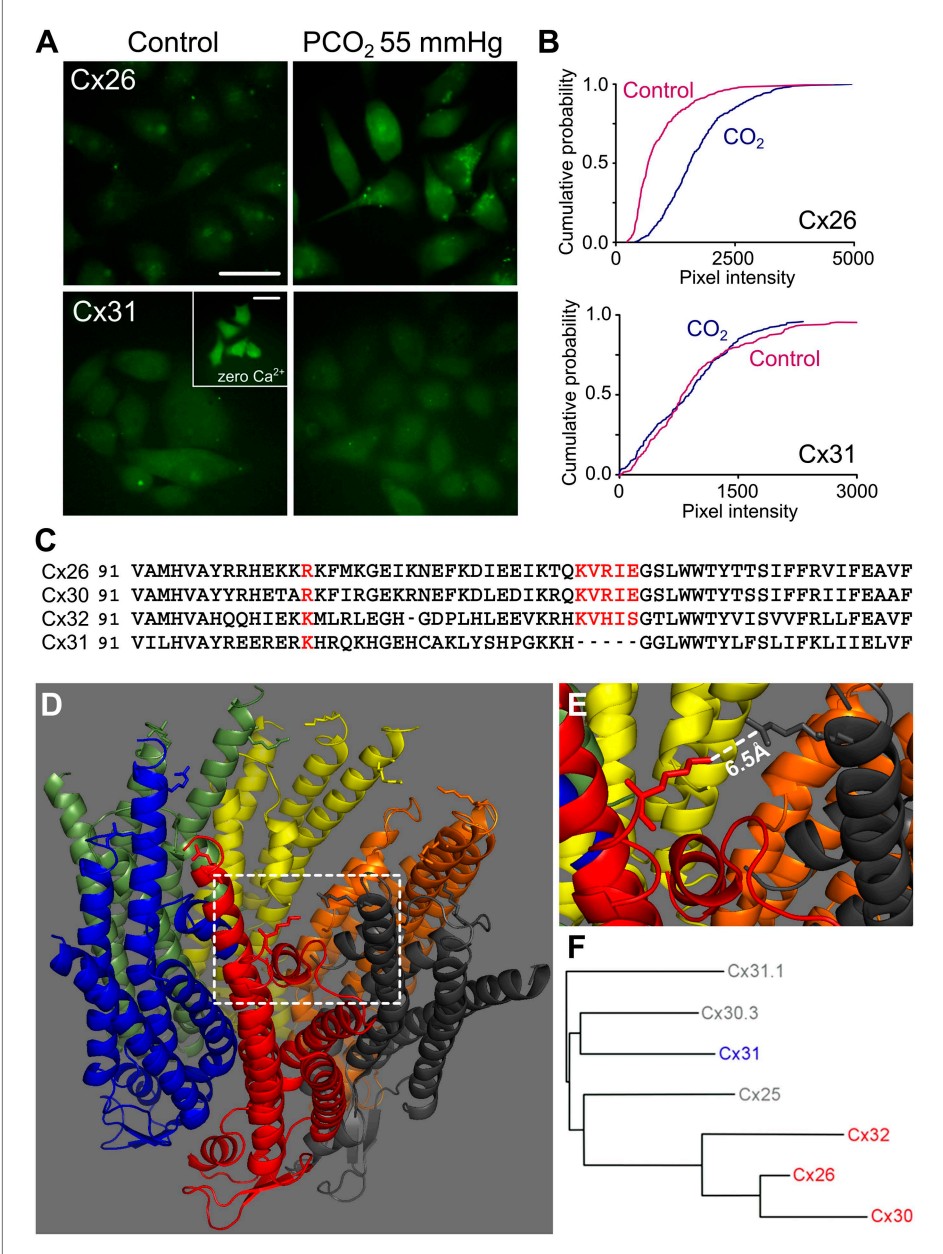

**Figure 1**. Identification of the motif in Cx26 that imparts $CO_2$ sensitivity. (**A**) Dye loading assay demonstrates $CO_2$-dependent loading of carboxyfluorescein into HeLa cells expressing Cx26, but not into those expressing Cx31. The inset in Cx31 shows that these hemichannels are expressed and functional in the membrane by utilizing a zero $Ca^{2+}$ stimulus to open them and allow dye loading. Scale bars 40 μm. (**B**) Cumulative probability plots of pixel intensity in the control and following exposure to $PCO_2$ of 55 mmHg. Each curve is comprises the measurements of mean pixel intensity for at least 40 cells. (**C**) Sequences (from mouse) for Cx26, 30, 32 and 31 to show K125 and four following amino acids that are present in Cx26, Cx30 and Cx32, but absent from Cx31. R104 in Cx26 and 30, and K104 in Cx32 and Cx31 are also highlighted. Accession numbers: Cx26, NP_032151; Cx30, AAH13811; Cx32, AAH26833; and Cx31, NP_001153484. (**D**) The structure of Cx26 drawn from the 2zw3 PDB file, cytoplasmic face of the channel upwards. On each subunit K125 and R104 are drawn. (**E**) Detail from the structure of Cx26 (dashed square) showing the orientation of K125 (red) towards R104 (dark grey). The short distance between the two amino acid side chains suggests that this gap could be bridged by carbamylation by $CO_2$ of K125 and a subsequent salt bridge with R104. (**F**) Phylogenetic tree showing relationship between Cx26 and other β connexins. The three $CO_2$ sensitive connexins are very closely related to each other while Cx31 is more distant.

*Figure 1. Continued on next page*

*Figure 1. Continued*

The following figure supplements are available for figure 1:

**Figure supplement 1**. Expression of Cx26 in HeLa cells imparts sensitivity to $CO_2$.

**Figure supplement 2**. Expression of connexin variants in HeLa cells.

lysine). We therefore made mutations that individually disrupted both of these functions: K125R to prevent carbamylation, and R104A to disrupt formation of the salt bridge. Neither Cx26[K125R] nor Cx26[R104A] exhibited sensitivity to $CO_2$ sensitivity. Nevertheless the positive controls demonstrated the presence of functional mutant hemichannels in the expressing HeLa cells (*Figure 3A*, *Figure 3—figure supplement 1*).

## Engineering an analogue of the carbamylated lysine into Cx26 makes it constitutively open

To test further our prediction that the carbamylated K125 forms a salt bridge with R104, we made the mutation K125E in Cx26. The insertion of glutamate in place of the lysine has the potential to act as an analogue of the carbamylated K125. If our hypothesis is correct, this mutated channel should be constitutively open, as the carboxyl group of the E125 should be able to form a salt bridge with R104. We found that HeLa cells expressing Cx26[K125E] readily loaded with dye under control conditions and exhibited no sensitivity to $CO_2$ (*Figure 4*). The Cx26[K125E]-expressing HeLa cells showed much greater loading under control conditions than parental HeLa cells (*Figure 4*, *Figure 4—source data 1*). To further confirm that the constitutive dye loading occurred via the misexpressed connexin, we

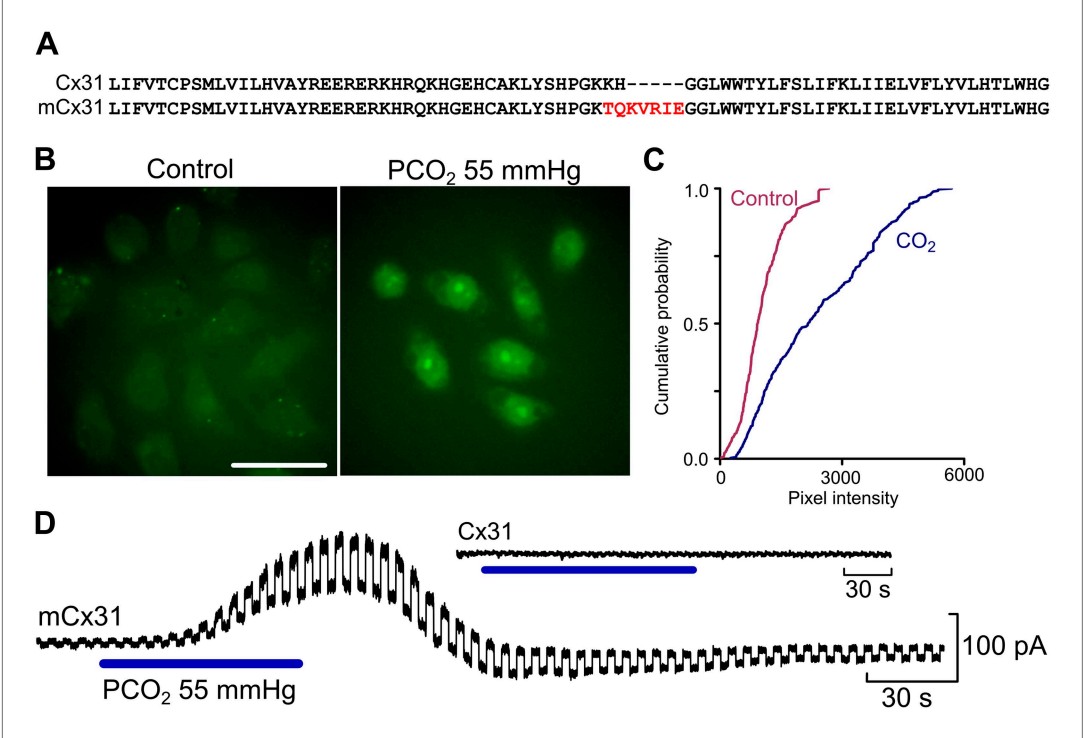

**Figure 2**. Insertion of the identified motif into Cx31 creates a $CO_2$-sensitive hemichannel. (**A**) Comparison of the WT and mutated Cx31 amino acid sequence to show the insertion of the K125 and surrounding residues. (**B**) The dye loading assay demonstrates gain of $CO_2$-sensitivity in mCx31. Scale bar 40 µm. (**C**) Cumulative probability of mean pixel density of 40 cells in five independent replications. (**D**) Whole cell patch clamp recordings from HeLa cells expressing mCx31 and Cx31. Recordings were performed under voltage clamp at a holding potential of −50 mV with a constant 10 mV step to assess whole cell conductance. The cells expressing mCx31 show a clear conductance change on exposure to a change in PCO₂, whereas cells expressing wild type Cx31 showed no such change (inset).

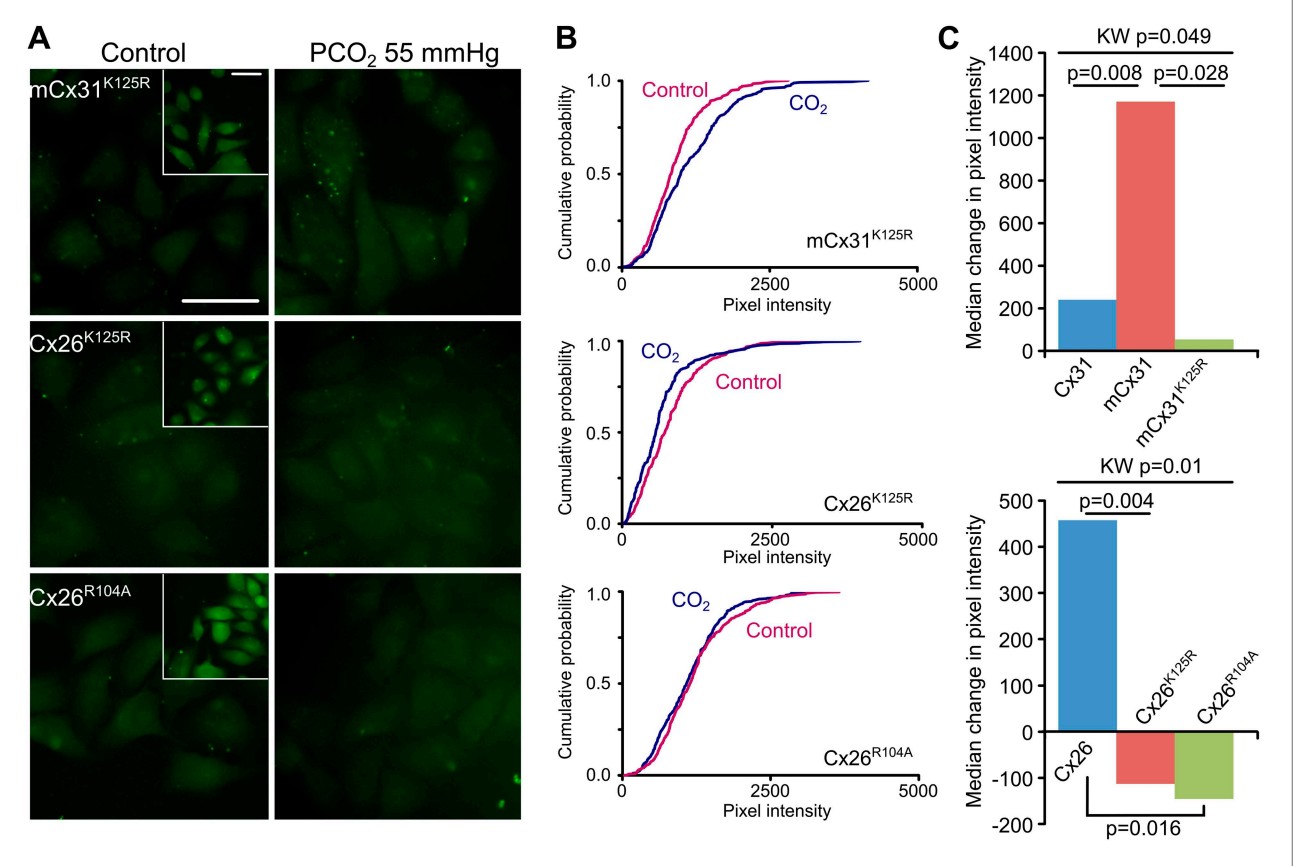

**Figure 3**. K125 and R104 are essential residues for $CO_2$ sensitivity. (**A**) Insertion of the motif (*Figure 2A*) from Cx26 but with the mutation K125R into Cx31 (mCx31$^{K125R}$) does not give a gain of $CO_2$ sensitivity indicating that this is an essential residue. Introducing the mutations K125R or R104A into Cx26 destroys the $CO_2$-sensitivity of Cx26. Insets show the zero $Ca^{2+}$ positive controls to demonstrate the presence of functional hemichannels in the cells. Scale bars, 40 µm. (**B**) Cumulative probability distributions demonstrate that none of these mutant channels are sensitive to $CO_2$. (**C**) Summary data demonstrating: gain of function in the mCx31 hemichannel and subsequent loss in mCx31$^{K125R}$; and loss of function in the Cx26$^{K125R}$ and Cx26$^{R104A}$ mutants. The graphs shown the median of the median change in pixel intensity from five independent replications for each type of hemichannel. KW: Kruskal-Wallis ANOVA, pairwise comparisons by the Mann-Whitney U-test.

The following source data and figure supplements are available for figure 3:

**Source data 1**. Median differences in pixel intensity between $CO_2$ and control dye loading experiments for the various connexin hemichannel variants in the histograms of *Figure 3C* and statistical analysis: Kruskal-Wallis anova, pairwise Mann-Whitney tests and false discovery rate procedure.

**Figure supplement 1**. All connexin variants form functional hemichannels capable of opening in response to zero $Ca^{2+}$.

demonstrated that carbenoxolone (100 µM) completely blocked $CO_2$-dependent dye loading in HeLa cells expressing Cx26$^{K125E}$ (*Figure 4*, *Figure 4—source data 1*).

Reasoning that if bridge formation between subunits was key to opening the hemichannel, we also made the further mutation R104E. In the mutated channel E104 has the potential to form a salt bridge in the reverse direction with K125 and we predicted that if this were to happen such a mutant hemichannel should also be constitutively open. We found that HeLa cells expressing Cx26$^{R104E}$ did indeed load with dye under control conditions and that this enhanced dye loading was blocked with carbenoxolone (*Figure 5*, *Figure 5—source data 1*).

## Elastic network model of Cx26 shows that carbamylation leads to hemichannel opening

Although our experimental data point to the importance of carbamylation of K125 and the formation of a salt bridge to R104 in the adjacent subunit, it is not clear how this would lead to opening of the

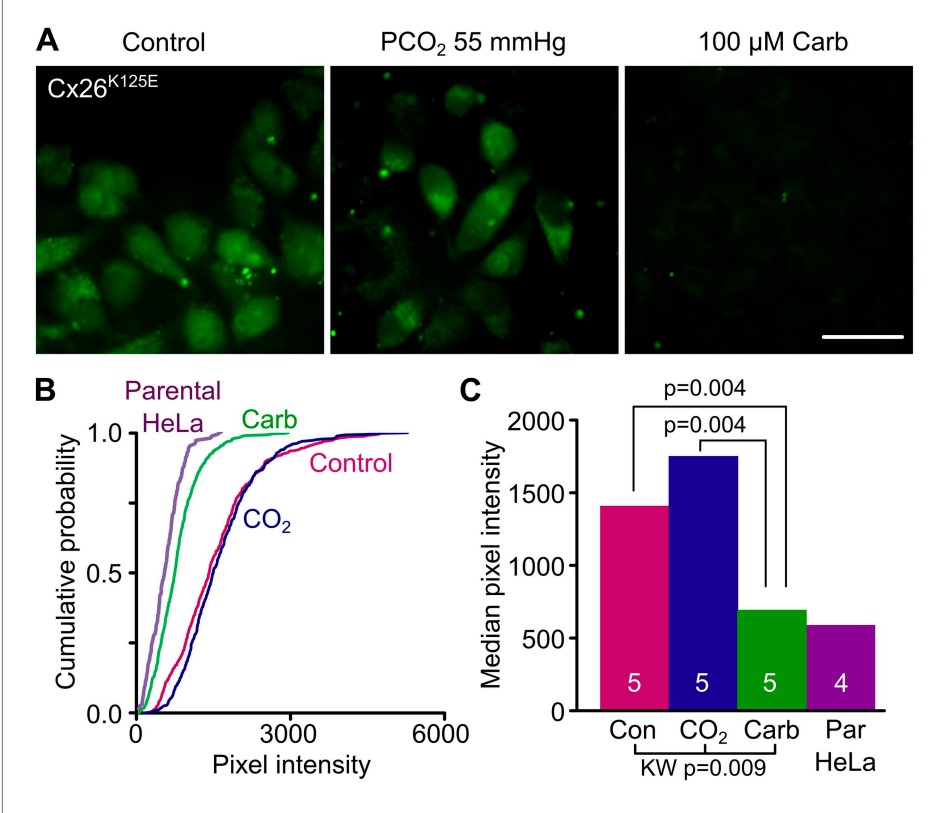

**Figure 4**. Engineering an analogue of the carbamylated lysine residue, Cx26[K125E], creates a constitutively open hemichannel that no longer responds to $CO_2$. (**A**) HeLa cells expressing Cx26[K125E] readily load with dye under control conditions. Increasing the $PCO_2$ does not give a further increase in dye loading. This dye loading was blocked by 100 µM carbenoxolone (Carb), indicating that it occurred through the heterologously expressed connexin. Scale bar 40 µm. (**B**) Cumulative probability plots comparing the median pixel intensities of at least 40 cells per experiment and five independent repetitions for the control, hypercapnic and carbenoxolone treatments with that of parental HeLa cells (four independent repetitions). (**C**) Summary data showing the median of the median pixel intensity for the three conditions for Cx26[K125E] and the background loading for parental HeLa cells. Pairwise comparisons by the Mann-Whitney U-test; KW Kruskall-Wallis Anova. Neither the difference between control and $CO_2$ nor the difference between Cx26[K125E] treated with carbenoxolone and parental HeLa cells is significant.

The following source data are available for figure 4:

**Source data 1**. Median pixel intensity values for histogram in *Figure 4C* and statistical analysis: Kruskal-Wallis anova and pairwise Mann-Whitney tests.

Cx26 hemichannel. Course-grained modelling reduces protein atomistic complexity for more efficient computational studies of harmonic protein dynamics and is particularly suited to examining hemichannel opening over millisecond time scales (*Sherwood et al., 2008*). We therefore built coarse-grained elastic network models (ENM) to gain insight into the mechanism by which $CO_2$ maintains Cx26 in the open state. In an ENM the Cα-atom co-ordinates of an atomic resolution structure are used to represent a protein structure. The total global protein motion within the ENM consists of a defined number of modes, each of a characteristic frequency and representing a particular harmonic motion within the protein. ENMs are known to reproduce the global low frequency modes of protein motion well in comparison to experimental data (*Delarue and Sanejouand, 2002*; *Valadie et al., 2003*). We used the co-ordinates from a high-resolution crystal structure to construct an ENM (*Tirion, 1996*) for the Cx26 hexamer in the unbound and $CO_2$-bound states. $CO_2$ was represented in the ENM by the inclusion of six additional Hookean springs between residues K125 and R104 of neighbouring monomers (*Figure 6A*).

Analysis of the model revealed that in the absence of $CO_2$ the lowest frequency mode (mode 1) represented an opening/closing motion that was able to fully occlude the hemichannel central pore

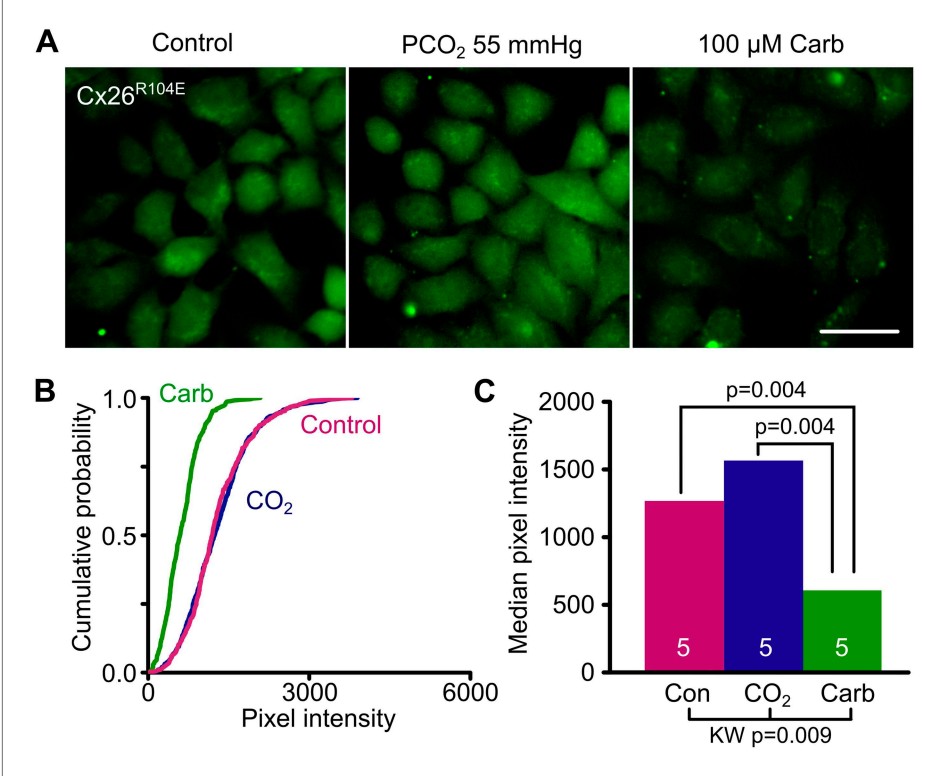

**Figure 5**. Bridging in the reverse direction: the mutation R104E forms a salt bridge with K125 in Cx26$^{R104E}$ to create a constitutively open hemichannel that no longer responds to $CO_2$. (**A**) HeLa cells expressing Cx26$^{R104E}$ readily load with dye under control conditions. Increasing the $PCO_2$ does not give a further increase in dye loading. This dye loading was blocked by 100 µM carbenoxolone (Carb), indicating that it occurred through the heterologously expressed connexin. Scale bar 40 µm. (**B**) Cumulative probability plots comparing the median pixel intensities of at least 40 cells per experiment and five independent repetition for the control, hypercapnic and carbenoxolone treatments. (**C**) Summary data showing the median of the median pixel intensity for the three conditions for Cx26$^{R104E}$. Pairwise comparisons by the Mann-Whitney U-test; KW Kruskall-Wallis Anova. The difference between control and $CO_2$ is not significant. DOI: 10.7554/eLife.01213.012

The following source data are available for figure 5:

**Source data 1**. Median pixel intensity values for histogram in **Figure 5C** and statistical analysis: Kruskal-Wallis anova and pairwise Mann-Whitney tests.

(**Video 1**). Addition of springs representing $CO_2$-binding to the ENM restricted the closing motions of the hemichannel and thus connexin 26 was maintained in the open state (**Video 2**). We examined the overlap in the ordering of the modes in the unbound and $CO_2$ bound states to gain insight into how this occurs. A significant reordering of the lowest frequency modes to higher frequencies was observed in the presence of $CO_2$ rather than the removal of any modes from the total protein motion (**Figure 6B**). Mode 1, the lowest frequency mode that represents the opening/closing mode, represented about 40% of the total protein motion in the absence of $CO_2$. In the presence of $CO_2$ this closing mode is reordered through a change in its frequency as mode 9, which accounts for only about 2% of the total motion (**Figure 6B,C**). $CO_2$ therefore opens Cx26 through a reordering of the normal modes of global protein motion such that the normal closing motion of Cx26 no longer significantly contributes to the total motion of the hemichannel. We infer from this that the carbamate bridge formed between Cx26 monomers represents a constraining force that hinders hemichannel closure.

## Discussion

Evidence from six different experimental tests supports our hypothesis that $CO_2$ forms a carbamate bridge between K125 and R104 on the adjacent subunit to open the Cx26 hemichannel. Firstly, we demonstrated the sufficiency of the carbamylation motif to confer $CO_2$ sensitivity by inserting it into a $CO_2$-insensitive

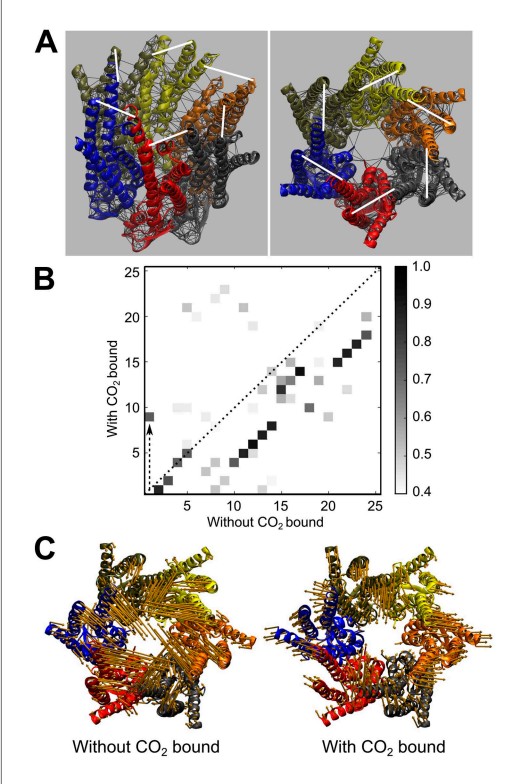

**Figure 6**. Elastic network model (ENM) of Cx26 demonstrating that $CO_2$ binding restricts the motion of the hemichannel and biases it to the open state. (**A**) Left, diagram of Cx26 from the 2zw3 structure, indicating the ENM (black lines) superimposed on the tertiary structure of Cx26 and showing the position of the hookean springs (white lines) introduced to simulate binding of $CO_2$ to K125 and bridging to R104. Right, ENM of Cx26 seen end on from the cytoplasmic side of the channel showing the six springs (white lines) that represent $CO_2$ binding. (**B**) Frequency modes of channel motion plotted for $CO_2$ bound against those without $CO_2$ bound. The grey scale on the right indicates the similarity of the modes between the $CO_2$ bound and unbound states. Note that there is a high degree of similarity between most modes in the bound and unbound state, indicating that binding of $CO_2$ reorders the modes of motion. In the graph, the modes that fall on the dotted line (x = y) have not changed between the two states. Mode 1 without $CO_2$ bound (closing of hemichannel) moves to Mode 9 with $CO_2$ bound (dashed upward arrow) indicating that it contributes much less to the total channel motions when $CO_2$ is bound. Most of the other modes fall below the dotted line, indicating that they become more frequent when $CO_2$ is bound. (**C**) Vectors indicating the Mode 1 motions of the α helices without $CO_2$ bound (left) and with $CO_2$ bound (right). The binding of $CO_2$ and creation of the carbamylation bridge between subunits greatly restricts hemichannel motion.

connexin, Cx31. Secondly, we showed that K125 of the carbamylation motif was essential for this motif to confer $CO_2$ sensitivity on Cx31. Thirdly and fourthly, we demonstrated that the mutations K125R and R104A in Cx26 (to prevent carbamate bridging at either end of the bridge) destroyed the $CO_2$ sensitivity of this connexin. Fifthly, by exploiting glutamate as an analogue of the carbamylated K125 (in Cx26[K125E]), we demonstrated a gain of function—Cx26[K125E] was constitutively open, yet had lost sensitivity to $CO_2$. Finally, we further tested the bridging concept by demonstrating that the bridge is in effect bidirectional: the mutated hemichannel Cx26[R104E], in which E104 can bridge to K125 in the reverse direction, was also constitutively open, but had no sensitivity to $CO_2$.

Although we have not directly demonstrated $CO_2$ binding to Cx26, our extensive testing of this hypothesis through selective mutations leads us to conclude that $CO_2$ interacts with Cx26 directly and that no other protein is required for $CO_2$ sensitivity. This interaction is most probably via carbamylation of K125. Interestingly, the mutations Cx26[K125E] and Cx26[K125R] can be considered respectively as open-state and closed-state analogues of the wild type channel. Collectively, our data strongly suggests that $CO_2$ binds to the intracellular surface of Cx26 and must therefore cross the membrane to reach this site. This could occur either direct diffusion through the membrane bilayer, potentially via Cx26 itself, or via other $CO_2$ permeable channels (***Boron et al., 2011***). Amongst its many other functions, Cx26 can therefore be regarded as a receptor for $CO_2$. Interestingly, this mechanism of modulation applies to both Cx30 and Cx32, which can both potentially form a carbamate bridge at equivalent residues to Cx26. In the case of Cx32 this would involve bridging to K104 rather than R104 (in Cx26 and 30). Cx26 can co-assemble with both Cx30 and Cx32 to form heteromeric hemichannels (***Forge et al., 2003***; ***Yum et al., 2007***). Our structural studies predict that, as Cx30 and Cx32 have K125 and either R104 or K104, carbamate bridges could form in such heteromeric hemichannels and that they should also therefore be $CO_2$-sensitive.

Carbamylation involves formation of a labile covalent bond between the carbon of $CO_2$ and a primary amine. For this to occur the amine must be in a restricted hydration space so that it is not protonated. Some examples of physiologically significant carbamylation are known. The carbamylation of the N-terminal amines of haemoglobin contributes to the Bohr effect (***Kilmartin and Rossi-Bernardi, 1971***), whereby the affinity of haemoglobin for $O_2$ is reduced in the presence of elevated $CO_2$. However in mammalian systems

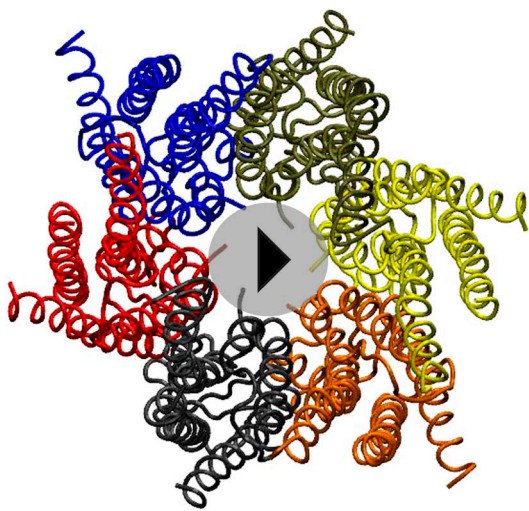

**Video 1**. Hemichannel mode 1 motions in absence of $CO_2$.

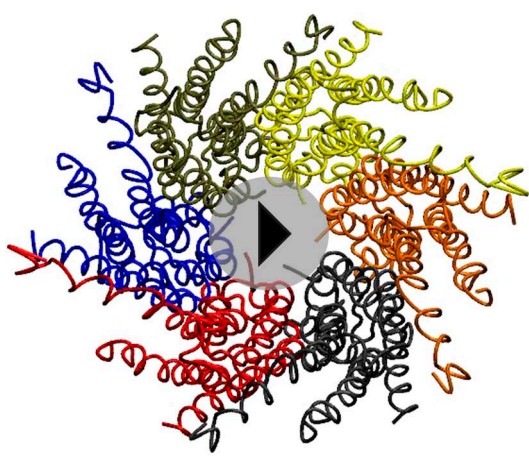

**Video 2**. Hemichannel mode 1 motions in presence of $CO_2$.

no other examples of carbamylation by $CO_2$ have been described. In C3 photosynthetic plants, the enzyme RuBisCo, that participates in the Calvin cycle and carbon fixation is activated by carbamylation of a specific lysine residue (*Lundqvist and Schneider, 1991*). Several microbial enzymes are also carbamylated (*Maveyraud et al., 2000*; *Golemi et al., 2001*; *Young et al., 2008*).

Despite this precedent, the functional significance of $CO_2$-carbamylation and its potential as a transduction principle for the measurement of $CO_2$ has been almost completely overlooked in vertebrate physiology. The mechanism of formation of a salt bridge between a carbamylated lysine and an appropriately oriented arginine on the neighbouring subunit is a unique mechanism for modulation of an ion channel and establishes carbamylation as a mechanistic basis for the direct signalling of $PCO_2$ in mammalian physiology. This carbamylation of a lysine to transduce the concentration of $CO_2$ into a biological signal is somewhat equivalent to the nitrosylation of a cysteine residue by NO/nitrite. It establishes a $CO_2$-dependent signalling paradigm in which the concentration of $CO_2$ is signalled by ATP release via Cx26 from the chemosensory cell and consequent activation of neighbouring cells, or potentially by a $Ca^{2+}$ influx through the Cx26 hemichannel (*Fiori et al., 2012*) to initiate a $Ca^{2+}$ wave within the chemosensory cell itself and further $Ca^{2+}$-dependent signalling processes.

## Materials and methods

### Hemichannel expression and mutagenesis

All connexin genes except Cx26[R104A], Cx26[K125E] and Cx26[R104E] were synthesised by Genscript USA and subcloned into the pCAG-GS mCherry vector. The sequence for wild type Cx26 and Cx31 genes were respectively take from accession numbers NM_001004099.1 and NM_019240.1. To produce Cx26[R104A], Cx26[K125E] and Cx26[R104E] site directed mutagenesis was performed using Quikchange II site directed mutagenesis kit. All wild type and mutant genes were sequenced to verify that the correct sequence was present. Hela cells were maintained in Dulbecco's modified Eagle's medium (DMEM) (Sigma-Aldrich Company Ltd, Gillingham, UK), 10% FCS (Biosera Europe, Labtech International Ltd, Uckfield, UK), 1:1000 pen/strep and supplemented with 3 mM $CaCl_2$. Cells were grown in a humidified 5% $CO_2$ incubator at 37°C. The connexin proteins were expressed via transient transfection. Cells were plated in six-well plates at $1 \times 10^5$ cells per well for Cx26 and its mutants and $5 \times 10^4$ cells per well for Cx31 and its mutants. Following the GeneJuice transfection reagent (Merck-Millipore UK, Merck KGaA, Darmstadt, Germany) user protocol, cells were transfected with 1 μg of the appropriate DNA. Experiments were performed when the connexin proteins had reached the cell membrane. This was found to be approximately 2 days for Cx26 and its mutants and approximately 3 days for Cx31 and its mutants.

### Solutions used
#### Standard artificial cerebrospinal fluid (aCSF, normocapnic)
124 mM NaCl, 3 mM KCl, 2 mM $CaCl_2$, 26 mM $NaHCO_3$, 1.25 mM $NaH_2PO_4$, 1 mM $MgSO_4$, 10 mM D-glucose saturated with 95% $O_2$/5% $CO_2$, pH 7.5, $PCO_2$ 35 mmHg.

## 50 mM $HCO_3^-$ aCSF (isohydric hypercapnic)

100 mM NaCl, 3 mM KCl, 2 mM $CaCl_2$, 50 mM $NaHCO_3$, 1.25 mM $NaH_2PO_4$, 1 mM $MgSO_4$, 10 mM D-glucose, saturated with 9% $CO_2$ (with the balance being $O_2$) to give a pH of 7.5 and a $PCO_2$ of 55 mmHg respectively.

### Dye loading assay and image analysis

Connexin expressing HeLa cells were plated on cover slips. A coverslip was placed in a small flow chamber and the cells were exposed to either: control aCSF with 200 μM carboxyfluorescein for 10 min; isohydric hypercapnic aCSF with 200 μM carboxyfluorescein for 10 min; or zero $Ca^{2+}$, 1 mM EGTA-containing aCSF plus 200 μM carboxyfluorescein for 10 min. This was followed by control aCSF plus 200 μM carboxyfluorescein for 5 min and then thorough washing for 30 min with control aCSF. These protocols are summarized in *Figure 7*.

The cells were then imaged by epifluorescence (Scientifica Slice Scope (Scientifica Ltd, Uckfield, UK), Cairn Research OptoLED illumination (Cairn Research Limited, Faversham, UK), 60x water Olympus immersion objective, NA 1.0 (Scientifica), Hamamatsu ImageEM EMCCD camera (Hamamatsu Photonics K.K., Japan), Metafluor software (Cairn Research)). Using ImageJ, the extent of dye loading was measured by drawing a region of interest (ROI) around individual cells and calculating the mean pixel intensity for the ROI. The mean pixel intensity of the background fluorescence was also measured in a representative ROI, and this value was subtracted from the measures obtained from the cells. All of the images displayed in the figures reflect this procedure in that the mean intensity of the pixels in a representative background ROI has been subtracted from every pixel of the image. At least 40 cells were measured in each condition, and the mean pixel intensities plotted as cumulative probability distributions.

For the dye loading experiments, the median pixel intensities of the control and $CO_2$ dye loading conditions (minimum of five independent repetitions) were compared by a Kruskal Wallace ANOVA and pairwise comparions by the Mann-Whitney test. The false discovery rate procedure (*Curran-Everett, 2000*) was used to determine whether the multiple pairwise comparisons remained significant.

### Patch clamp recordings

Cover slips containing non-confluent cells were placed into a perfusion chamber at 28°C in sterile filtered standard aCSF. Standard patch clamp techniques were used to make whole-cell recordings. The intracellular fluid in the patch pipette contained: K-gluconate 120 mM, CsCl 10 mM, TEACl 10 mM, EGTA 10 mM, ATP 3 mM, $MgCl_2$ 1 mM, $CaCl_2$ 1 mM, sterile filtered, pH adjusted to 7.2 with KOH. All whole-cell recordings were performed at a holding potential of −40 mV with regular steps of 5 s to −50 mV to assess whole-cell conductance.

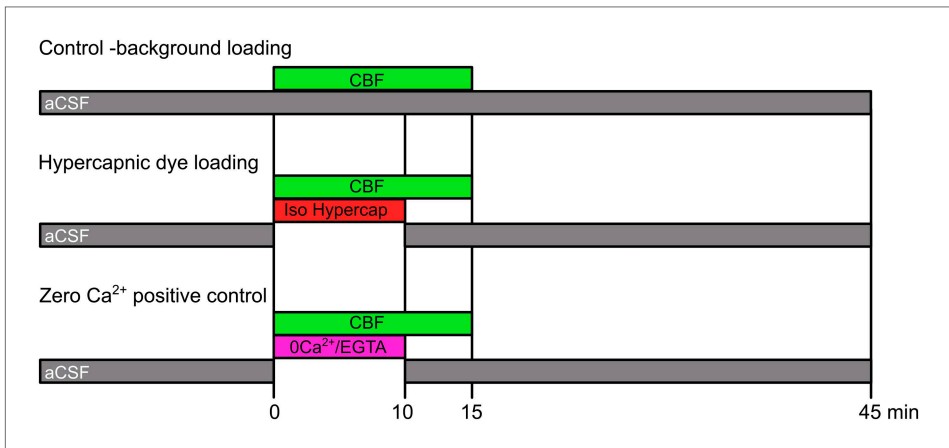

**Figure 7**. Dye loading protocols. The control background loading tests for any potential $CO_2$-insensitive pathways of dye loading that are constitutively active in the HeLa cells. Hypercapnic dye loading uses the 50 mM $HCO_3^-$ aCSF to test $CO_2$-sensitive loading under conditions of isohydric hypercapnia ($PCO_2$ 55 mmHg). The zero $Ca^{2+}$ positive control tests for the presence of functional hemichannels in those cases where the misexpressed hemichannels exhibit no sensitivity to $CO_2$.

## Elastic network model–course-grained simulations

Elastic network model (ENM) simulations were performed based on its regular implementation using pdb file 2ZW3, where all the Cα atoms in the protein within a given cut-off radius are joined with simple Hookean spring (*Tirion, 1996*; *Rodgers et al., 2013a*). The spring constants were set to a constant value of 1 kcal mol$^{-1}$ Å$^{-2}$ with a cut-off radius of 8 Å. The presence of $CO_2$ molecules were represented in the ENM by the inclusion of an additional Hookean spring between residues K125 and R104 of each set of neighbouring monomers (*Rodgers et al., 2013b*). The first six modes, that is the lowest frequency modes, represent the solid body translational and rotational motions of the protein and are thus ignored from the analysis.

## Additional information

### Funding

| Funder | Grant reference number | Author |
|---|---|---|
| Medical Research Council | G1001259 | Nicholas Dale |
| Biotechnology and Biological Sciences Research Council | | Louise Meigh |
| Engineering and Physical Sciences Research Council | EP/H051759/1 | Thomas L Rodgers, Martin J Cann |

The funders had no role in study design, data collection and interpretation, or the decision to submit the work for publication.

### Author contributions

LM, ND, Conception and design, Acquisition of data, Analysis and interpretation of data, Drafting or revising the article; SAG, TLR, Acquisition of data, Analysis and interpretation of data; MJC, Acquisition of data, Analysis and interpretation of data, Drafting or revising the article; DIR, Conception and design, Analysis and interpretation of data, Drafting or revising the article

## Additional files

### Supplementary files

• Supplementary file 1. Conductance changes source data. Raw values for whole cell conductance changes (nS) in response to an isohydric $CO_2$ challenge (PCO$_2$ 55 mmHg) in Cx31 and mCx31 expressing HeLa cells.

### Major dataset

The following previously published dataset was used:

| Author(s) | Year | Dataset title | Dataset ID and/or URL | Database, license, and accessibility information |
|---|---|---|---|---|
| Maeda S, Nakagawa S, Suga M, Yamashita E, Oshima A, Fujiyoshi Y, Tsukihara T | 2009 | Structure of the connexin 26 gap junction channel at 3.5 A resolution | 2ZW3; http://www.rcsb.org/pdb/explore/explore.do?structureId=2ZW3 | Publicly available at RCSB Protein Data Bank (http://www.rcsb.org). |

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
