## [Decision Letter]

Thank you for sending your work entitled “CO_2_ directly modulates connexin 26 by formation of carbamate bridges between subunits” for consideration at *eLife*. Your article has been favorably evaluated by a Senior editor and 2 reviewers, along with a member of our Board of Reviewing Editors.

The following individuals responsible for the peer review of your submission have agreed to reveal their identity: Michael Marletta (Senior editor); Richard Aldrich (Reviewing editor); Juan Saez (peer reviewer).

The Reviewing editor and the reviewers discussed their comments before we reached this decision, and the Reviewing editor has assembled the following comments to help you prepare a revised submission.

The observations in the manuscript by Meigh et al. are highly novel and represent a new way in which we can look at the body's ability to sense CO_2_. The involvement of connexin hemichannels in this process is particularly novel. The data presented to implicate direct carbamylation of the connexin protein (at K125 in Cx26) are compelling, given that CO_2_ induced opening of hemichannels is eliminated using mutants at this site, and can be introduced into a non-CO_2_ sensitive Cx31 by inserting the target sequence.

However, the authors conclude strongly that: (1) the carbamylation of Cx26 (and presumably the other CO_2_ sensitive connexins) leads directly to opening of hemichannels, and that, (2) this is caused by formation of a salt bridge with R104. The data for both these claims is currently inadequate.

With respect to (1), the authors do show that the increased membrane conductance and dye uptake are dependent on Cx expression. Thus, Cx hemichannels are a likely candidate to explain the leak, but other channels like Pannexins, or more recently, the CALHM channels, could similarly cause this conductance and dye permeability increase. Ca^2+^ block of this conductance is shown, but this is not very specific, and the authors do not show that it is reversible.

More definitive evidence that connexin hemichannels cause the leak is necessary. This should be easy to obtain by simply doing some single channel patch recordings, where Cx hemichannels are readily distinguished from Pnx1 channels. This would also allow the authors to determine if the changes in conductance really reflect enhanced open probability (as they conclude in the absence of evidence), or possibly as a result of enhanced trafficking or other changes. Different pharmacological blockers like La^+++^ that are specific for connexins could also be used to distinguish connexin hemichannels from other candidates.

With respect to (2), this is based on structural models of Cx26, at regions that are not well resolved in the original diffraction pattern. It is supported by one R104A mutant that disrupts CO_2_ gating. But mutants that ablate function are not very instructive, as this could be for many reasons. The authors could also test an R104K mutant that one might expect to retain at least partial function, if it is dependent on a salt bridge as proposed by the authors.

Carbamylation to control activity is well known. Rubisco is the best example but there are others such the beta lactamases as cited by the authors. This is mentioned in the Discussion. Carbamylation is the central theme of this paper and must be initially brought up in the Introduction.

In citing their past work where connexins 30 and 32 open in response to CO_2_ with constant pH, that constant pH is extracellular. Admittedly the bulk evidence supports a direct role of pCO_2_ and not a pH change with the increase in CO_2_, these experiments do not incisively rule out an intracellular pH effect.

In the opening of the discussion the authors state:

“Our analysis has demonstrated that CO_2_ binds to the intracellular surface of Cx26 and must therefore diffuse through the membrane to reach this site.”

Direct binding was not demonstrated. This is the most serious weakness in the paper. The experiments as designed are excellent but the authors stop short of the most critical molecular detail.

The definitive proof of a carbamoylated connexin structure is no doubt what the authors would like and so would we. ^14^C-CO_2_ binding in WT and mutants should be clear. And easy. The authors need to show binding with ^14^CO_2_ and/or the unique NMR signal generated with ^13^CO_2_.

Together, the electrophysiological and binding experiments would allow the authors to be more secure in their conclusions, which at this point are more speculative than presented in the manuscript.

---

## [Author Response]

*1) The identity of the hemichannel underlying the conductance change and dye permeation – in particular reassurance that this is due to the misexpression of Cx26 rather than some other endogenous conductance (such as pannexin1 and calhm1). There was also a request for single channel recordings*.

We published data to this effect in Huckstepp et al. 2010 J Physiol 588, 3901 and Huckstepp et al. 2010, J Physiol 588, 3921. In the first of these papers we demonstrated that CO_2_-dependent ATP release from the medulla was blocked by a number of agents active at connexin hemichannels (carbenoxolone at high concentration, proadifen, NPPB and Co^2+^), but not by agents selective for pannexin-1 (carbenoxolone at low concentration and probenecid). In the second of these papers we demonstrated that the whole cell conductance change in responses to changes in PCO_2_ were accompanied an increase in current noise (indicative of increased channel gating) and also performed single channel recordings in isolated membrane patches from Cx26-epxressing HeLa cells in the inside-out and outside-out configurations. These recordings demonstrated that increasing and decreasing PCO_2_ respectively increased and decreased single channel gating in both configurations of the isolated patches at a membrane potential of -40 mV (which would by itself rule out panx-1, as this hemichannel requires membrane depolarization to open).

However at the time we were unaware of the existence of calhm1, so did not specifically test whether this hemichannel could be involved. We therefore provide additional data, in a new supplement to Figure 1, to demonstrate the block of CO_2_-dependent dye loading by carbenoxolone (not active at calhm1, [27] Nature doi: 10.1038/nature11906), and a corresponding lack of effect of probenecid (to repeat earlier findings ruling out Panx-1) and ruthenium red (active at calhm1). These data show that only carbenoxolone prevents the CO_2_-dependent dye loading of the HeLa cells and thus rule out these alternative possibilities. We have also made more complete reference to our earlier work in the initial section of the results to address the issues of hemichannel identity and single channel conductances.

We also thought it would be helpful to show in and additional supplement to Figure 1 examples of the mCherry expression patterns for selected mutant and WT connexins.

*2) Direct binding of CO*_*2*_
*to the hemichannel has not been demonstrated, and the evidence supporting the bridge is weak*.

It is true that we have not directly shown the binding of CO_2_ to Cx26, and the demonstration of such would be highly desirable. Unfortunately we do not think that either alternative proposed by the reviewers is practicable owing to the large amount of protein required.

^14^CO_2_ binding typically requires 80 nmol of protein per single technical replicate (J Biol Chem [1979] 254, 5599-601) corresponding to just over 2 mg of Cx26 released from membranes (typically up to 10% of total expressed and purified Cx26 – J Cell Biol [1991] 115, 141-50). This corresponds to about 20 mg of protein for 3 biological replicates with 3 technical replicates each. The total protein required would be considerably more with method development. Given that HeLa cells express only low amounts of connexin protein, this approach is therefore unfeasible for Cx26 with the expression systems we have currently available to us. Similarly, large amounts of protein are required for ^13^C-NMR (62.5 mg/ml protein BBRC [1983] 111, 544; 0.5 mM protein J Biol Chem (1976) 251, 477 and (1977) 252, 2234; 70 mg/ml protein Proc Natl Acad Sci [1979] 76, 673), even allowing for the increased sensitivity modern machines.

However, we have given further thought as to how we could provide further evidence to support both the concept of CO_2_ binding and the concept of the carbamate bridge that the reviewers correctly point out is supported by only a single mutation. We reasoned that we could in effect “engineer in” the binding of CO_2_ by replacing K125 with a glutamate residue. The carboxy side chain of glutamate has the potential to provide the same functionality as the carbamylated lysine and could thus form a salt bridge with R104. We would predict that such a channel (Cx26^K125E^) should be both constitutively open and no longer sensitive to CO_2_. We now present evidence that both these predictions are true and that Cx26^125E^ can be regarded as an analogue of the open, carbamylated hemichannel.

The suggestion that we test the mutant R104K is a good one. However in our gain of function mutant, mCx31, the other end of the bridge is K104 and the mutation R104K in Cx26 would give relatively little further information as we already know that a lysine at this position can substitute for the arginine. As an alternative approach, it struck us that we could provide further support for the idea of the bridge between K125 and R104 simply by reversing the direction of the bridge. The mutation R104E provides the required carboxy functionality from residue 104 and it should, on the basis of our hypothesis, be able to form a salt bridge in the reverse direction with K125 and thus also give a constitutively open hemichannel. We now provide evidence that Cx26^R104E^ is also constitutively open, and no longer sensitive to CO_2_.

We summarize these new data for Cx26^K125E^ and Cx26^R104E^ in Figures 4 and 5. We have amended the Discussion to the effect that we have not demonstrated direct binding of CO_2_, but that 6 experimental tests derived from the hypothesis of CO_2_ binding (gain of function in mCx31, loss of function in mCx31^K125R^, loss of function in Cx26^K125R^, loss of function in Cx26^R104A^, gain of function in Cx26^K125E^, and gain of function in Cx26^R104E^) provide results that strongly support this hypothesis.

*3) Carbamylation is the central theme of this paper and must be initially brought up in the Introduction*.

We now introduce the concept of carbamylation in the Introduction.

*4) Intracellular pH – a possible contributor*?

Our previous work in the 2010 papers leaves very little room for intracellular pH changes as the causation of the hemichannel gating or CO_2_-dependent ATP release via Cx26. Perhaps the strongest evidence from this previous work is that isolated membrane patches in both inside out or outside out configuration exhibit changes in channel gating under conditions of constant bath pH (Huckstepp et al. 2010, J Physiol 588, 3921). Our new data, especially the observation that K125E and R104E mutants are both constitutively open and lack CO_2_ sensitivity, makes the concept of pH changes underlying the CO_2_-dependent channel gating even less likely. We have opted not to add any discussion of this in the paper.